# Retrospective Analysis of Bone Substitute Material for Traumatic Long Bone Fractures: Sex-Specific Outcomes

**DOI:** 10.3390/ijms241814232

**Published:** 2023-09-18

**Authors:** Jonas Pawelke, Vithusha Vinayahalingam, Christian Heiss, Thaqif El Khassawna, Gero Knapp

**Affiliations:** 1Experimental Trauma Surgery, Faculty of Medicine, Justus Liebig University, 35392 Giessen, Germany; jonas.pawelke@med.uni-giessen.de (J.P.); vithusha.vinayahalingam@med.uni-giessen.de (V.V.); christian.heiss@chiru.med.uni-giessen.de (C.H.); thaqif.elkhassawna@chiru.med.uni-giessen.de (T.E.K.); 2Department of Trauma, Hand and Reconstructive Surgery, Faculty of Medicine, Justus Liebig University, Rudolf-Buchheim-Straße 8, 35392 Giessen, Germany

**Keywords:** gender medicine, sex, musculoskeletal system, bone substitutes, fractures, calcium phosphates, hydroxyapatites

## Abstract

Male patients often experience increased bone and muscle loss after traumatic fractures. This study aims to compare the treatment outcomes of male and female patients with large bone defects. A total of 345 trauma patients underwent surgery, with participants divided into two groups: one receiving bone substitute material (*BSM*) for augmented defects (*n* = 192) and the other without augmentation (empty defects = *ED*, *n* = 153). Outcome parameters were assessed among female (*n* = 184) and male (*n* = 161) patients. Descriptive statistics revealed no significant differences between male and female patients. Approximately one-half of the fractures resulted from high-energy trauma (*n* = 187). The *BSM* group experienced fewer complications (*p* = 0.004), including pseudarthrosis (*BSM*: *n* = 1, *ED*: *n* = 7; *p* = 0.02). Among female patients over 65, the incidence of pseudarthrosis was lower in the *BSM* group (*p* = 0.01), while younger females showed no significant differences (*p* = 0.4). Radiologically, we observed premature bone healing with subsequent harmonization. Post hoc power analysis demonstrated a power of 0.99. Augmenting bone defects, especially with bone substitute material, may reduce complications, including pseudarthrosis, in female patients. Additionally, this material accelerates bone healing. Further prospective studies are necessary for confirmation.

## 1. Introduction

The recognition of gender significance and the use of innovative bone graft substitutes in medical practice have grown in recent years. The increasing understanding of endocrinological differences between male and female patients has sparked scientific investigations and scholarly discussions. Focusing on gender-specific considerations and exploring alternative bone graft substitutes contribute to advancing medical knowledge and can greatly impact daily clinical decision making.

### 1.1. Preclinical and Experimental Studies on Gender Differences in Bone Healing

Previous in vitro experiments that compared sex differences in orthopedics used femur fractures in mice to examine the influence of systematic bone and muscle loss [1]. Both male and female mice showed an increase in cytokines IL-1 and IL-6, while TNF-α was only measured in males. Male subjects experienced greater regional bone mineral content and density loss, while females exhibited a trend toward increased muscle loss [1]. Additionally, aged female mice displayed increased resorption activity, larger osteoclasts, and elevated TRAP following alveolar extraction [2]. Middle-aged female osteotomized rats exhibited impaired bone healing [3]. However, male mice revealed more cartilage but less fibrous tissue in the fracture callus compared with females [4]. Studies also indicated faster bone healing with a larger fracture callus caused by lower estrogen receptor expression [3]. Female rats and mice had reduced bridging compared with males [5,6]. Building upon this evidence, a prospective pilot study demonstrated the role of sex hormones in improving bone healing in male rats [3].

### 1.2. Transferability of Examinations on Gender Differences in Orthopedics

Gender differences extend beyond traditional roles and behaviors, as multiple trials have shown their impact on the body. The beneficial effects of estrogens in preventing osteoporosis have been extensively documented, while reports on osteoporosis in male patients are less prevalent [5,7]. Furthermore, there has been considerable debate regarding the potential benefits of estrogens in modulating cytokine release and chemotaxis [8]. Studies dating back to 1975 and 1999 have explored the influence of sex on infections, finding a higher incidence of systematic infections in male patients [9,10]. A current study demonstrated increased surgical site infections (SSI) in male patients [11]. Subsequent research revealed that male patients are at higher risk of developing post-traumatic infections due to increased chemotaxis and cytokine levels [12,13]. A comparison of fracture outcomes in nearly 500,000 patients in Taiwan showed that male patients experienced more complications and higher infection rates. Additionally, the 30-day mortality rate was higher in males, although urinary tract infections decreased because of anatomical factors [14]. Studies on female patients, especially the elderly, highlighted the higher prevalence of fractures resulting from low-energy trauma in women compared with men [15,16]. Thus, the aging population and increased incidence of fractures in the elderly present a future challenge [17]. From the assumption of increased bone resorption activity following alveolar bone extraction in elderly mice, the bone healing process in female geriatric patients could be delayed [2]. Furthermore, elderly female patients exhibited decreased CFU-APs in the bone marrow essential for bone formation [18]. Recent studies have shown that bone substitute augmentation lowers the rate of nonunions in geriatric patients [19].

This study aimed to investigate gender-specific outcomes following traumatic bone fractures treated with and without bone augmentation surgery. A previous study assessed the use of demineralized freeze-dried bone allografts and found that the donors’ gender had no influence [20].

### 1.3. Purpose of this Study

In summary, sex differences in orthopedics and trauma surgery significantly impact the bone healing process [21]. Most research on sex-specific outcomes has been conducted in preclinical and in vitro trials, with limited clinical studies. This data gap hinders our understanding of how gender affects bone healing, with even less data available for transgender individuals. The primary objective of this study was to assess disparities in outcomes between genders, focusing on complications and bone healing after surgical intervention for long bone fractures resulting from trauma. Initially, these fractures were managed using standard best medical treatment, and then bone defect augmentation was applied to expedite the bone healing process in both male and female cohorts. Our research emphasized the importance of considering gender-specific factors in managing and treating trauma-related fractures and exploring potential interventions to enhance the healing process in both genders.

## 2. Results

### 2.1. Demographic Data

Between 2011 and 2018, 355 patients underwent trauma surgery in the level-one hospital. Ten patients were excluded because of pathological bone fractures, stress fractures, or missing data, leaving a total of 345 (100%) patients for evaluation.

Of these patients, 184 (53.3%) were female, and 161 (46.7%) were male. The average age was 55.8 ± 17.6 (range 15–91) years, with female patients having a higher average age of 63.0 ± 16.0 (range 18–91) compared with male patients’ average age of 47.5 ± 15.6 (range 15–83). Patients were divided into two age cohorts on the basis of the definition of geriatric traumas (>65 years) from previous trials, with most patients falling into the nongeriatric category 239 (69.3%). Female patients had a higher proportion of geriatric fractures (89, (48.4%)) and a lower proportion of non-geriatric fractures (95, (51.6%)) compared with male patients, who had 17 (10.6%) geriatric fractures and 144 (89.4%) nongeriatric fractures (Figure 1). These differences between genders in geriatric and nongeriatric patients were not statistically significant for both female (*p* = 1.0) and male cohorts (*p* = 0.5), indicating that outcomes in each gender group were comparable.

The left side of the body was affected in 201 (58.3%) cases, while the right was affected in 141 (40.9%) cases. Among all female patients, the left body side was affected in 105 (57.1%), and the right side was affected in 77 (41.8%) cases. In the male patient group, there was a nonsignificant difference between the left side, 96 cases (59.6%), and the right side, 64 cases (39.8%). Previous illness, as classified by the American Society of Anesthesiologists Physical Status Classification System (ASA classification), showed an accumulation at ASA Type 2 in 222 cases (64.3%). Fewer patients (66) were categorized as ASA 3 (19.1%), 45 as ASA 1 (13.0%), and 1 as ASA 4 (0.3%). Because of lost data, 11 patients (3.2%) could not be categorized (Table 1). The average Charlson Comorbidity Index (CCI) was 2.8, with a median of 55 cases (15.9%) with CCI 0. There were no differences between the treatment groups (*p* = 0.4), and there was no significant difference between the male and the female groups in terms of bone substitute augmentation and empty defect treatment cohorts (*p* > 0.05).

Bone substitute material (BSM) was employed in 192 cases (55.7%), while no material was added in 153 (44.3%) with empty defects (ED). Most fractures were treated with calcium phosphate-based bone replacement material. Specially, Calcibon^®^ (Zimmer Biomet Deutschland GmbH, Freiburg, Germany) was used in 137 cases (39.7%), while hydroxyapatite-based material Ostim^®^ (aap Biomaterials GmbH, Dieburg, Germany), was used in 55 patients (15.9%). Within the female patient cohort, there were 81 cases (44.0%) without *BSM* and 103 (56.0%) with the addition of alloplastic bone material, averaging 2.4 mL. In the male patient group, no difference was observed, with 72 patients (44.7%) undergoing ED treatment and 89 (55.3%) receiving BSM, averaging 2.4 mL of bone substitute. Furthermore, comparing the male and the female groups, no differences were observed in the use of bone substitutes (*p* > 0.05) (Figure 2). 

The average body mass index (BMI) was 26.8 ± 4.9 (range 17.1–44.4) kg/m^2^. In the female cohort, the average BMI was 26.7 ± 5.3 (range 17.1:44.4) kg/m^2^, showing a nonsignificant difference compared with male patients 27.1 ± 4.5 (range 17.2–44.1) kg/m^2^. Using the WHO classification of BMI and clinical obesity groups, 5 patients (1.4%) had a BMI less than 18.5 kg/m^2^, 129 (37.4%) had a normal weight of up to 24.9 kg/m^2^, and 127 (36.8%) had a BMI less than 29.9 kg/m^2^. Grade one obesity (BMI < 34.9) was observed in 44 patients (12.8%) patients, while grade two (BMI < 39.9) and grade three obesity (BMI > 40.0) were found in fewer cases, 20 (5.8%) and 6 (1.7%), respectively (Figure 3). Comparing the groups of female and male patients and their BSM and ED treatments, no significance was observed (*p* > 0.05).

The causes of trauma were divided into groups: 187 (54.2%) high-energy trauma, 148 (42.9%) low-energy trauma, and 8 cases (2.3%) where the cause was indeterminable because of the missing medical history. Further subdivision revealed accidents due to stumbling in 96 patients (27.8%), domestic falls in 46 (13.3%), and sport-related accidents in 46 cases (13.3%). Thirty-eight traffic accidents involving cars (11.0%), 21 motorcycles (6.1%), and 16 bicycles (4.6%) were also observed. Falls from a height of more than 2 m occurred in 42 cases (12.2%), while a stumbling fall on stairs caused 21 accidents (6.1%). Because of incomplete medical history, 14 traumas (4.1%) could not be definitively categorized.

Comparing all demographic data revealed no significant differences (*p* > 0.05).

### 2.2. Postsurgical Complications

As the most significant part of daily clinical practice, postsurgical complications were thoroughly assessed. Among all analyzed patients, complications were absent in 209 (60.6%), while some were observed in 136 patients (39.4%). Comparing complications between female and male patient groups, no statistically significant difference was found (*p* = 0.4). In the female patient group, no complications were reported in 115 cases (62.5%), while 69 patients (37.5%) experienced complications. In the male group, postoperative complications occurred in 67 fractures (41.6%) (Figure 4).

Given their clinical relevance, complications were categorized and initially compared among all groups, whether they received the bone defect filling or not. There were no statistically significant differences between females and males with respect to complications and comorbidities. The parameters examined included pseudarthrosis, necrosis, infection, delayed bone healing, soft tissue injuries, chronic regional pain syndrome (CRPS), inactivity osteoporosis, arthrosis, chondromalacia, secondary dislocation of the osteosynthesis, psychological illness, neurological diseases, and premature removal of the osteosynthesis (Table 2).

Upon subdividing the dataset into female and male patients, we observed certain significant advantages in specific groups. Female patients, for instance, exhibited a lower number of complications in the group with added bone graft material (30 (29.1%)) but a higher number of complications (39 (48.1%)) in the ED group (*p* = 0.004). To prevent one complication, the number of patients needed to be treated (NNT) was 5.3. Furthermore, one of the severe complications in orthopedic trauma surgery—pseudarthrosis—showed decreased cases when treating fractures with BSM (*p* = 0.02). Pseudarthrosis was observed in 7 cases (8.6%) in the female group without BSM, compared with 1 (1.0%) in fractures with BSM augmentation. To prevent one pseudarthrosis in the female patient group, the number of patients needed to treat was 13.2. For further specification, all patients were categorized by gender, age (less than 65 and over 65 years), and treatment (ED and BSM groups). Comparing these parameters indicated no differences in female patients under 65 (*p* = 0.4). However, female patients over 65 treated with ED demonstrated a higher number of patients (6 (15.4%)) who suffered one (2.0%) pseudarthrosis compared with the group of BSM augmentation (*p* = 0.01). There were no differences observed when comparing the male patient groups and both treatment options (*p* = 1.0, Table 3). The NNT to prevent one pseudarthrosis in the female group aged over 65 years was 7.5. In the follow-up examination, fewer chondral cartilage defects were noticed in the female patient group with bone defect filling (*p* = 0.05), with 16 cases in the ED group (19.8%) and 9 in the BSM group (8.7%). Other complications under evaluation did not exhibit significant differences (*p* > 0.05). Male patients showed fewer complications, with 11 complications when a bone substitute was used in the fractures (15.3%) compared with 37 in the empty defects group (*p* = 0.03) (51.4%).

### 2.3. Radiological Bone Healing Process

The bone healing process was assessed using the criteria established by Bohnhof et al., Freyschmidt et al., and Islam et al. [22,23,24]. The follow-up examinations were conducted immediately, 1.9 ± 1.2 days after surgery (range 0–7), and at several intervals afterward; at 24.3 ± 6.4 days (range 10–40), 50.9 ± 8.6 (range 40–70), 90.7 ± 15.7 (range 60–130), and 247.0 ± 71.7 days (range 135:365), with a long-term follow-up at 638.2 ± 307.7 (range 371–1986) days after the surgery. The first postsurgical examination revealed no differences in the fracture edge among female and male patients in both ED and the BSM groups (*p* = 0.6). However, significant differences between the treatment options emerged in subsequent follow-up examinations after 24, 52, 91, and 247 days for both female and male patient groups (*p* < 0.05). The long-term follow-up examination showed a convergence between the ED and BSM treatment groups for both female and male patients. Similar observations were made when evaluating the fracture gap-healing process. Comparing the articular surface between female and male patients revealed fewer pathological patterns in the groups that received bone defect augmentation (*p* < 0.05). The types of osteosynthesis material did not yield differences among the ED and BSM treatment groups (*p* > 0.05) (Table 4).

Calculating the post hoc power analysis for the Mann–Whitney test of two groups (n1 = 184 and n2 = 161) and the alpha error of 0.05, the trial showed a good test power of 0.99. The power analysis was calculated using the G*Power^®^ program [25].

## 3. Discussion

### 3.1. Summary of Findings

Long bone fractures, resulting from various traumatic and nontraumatic injuries, present challenges for orthopedic surgeons in achieving stable union and preventing complications. A popular approach to address these challenges is using BSM-based bone defect augmentation to provide additional structural support and aid bone healing. However, limited information is available on sex-specific differences in outcomes for male and female patients undergoing BSM augmentation. This retrospective case-control study investigated the treatment of bone defects using BSM in male and female patients with traumatic long bone fractures. The study has produced valuable insights into clinical and radiological outcomes following bone defect augmentation.

Considering their clinical significance, postsurgical complications were thoroughly assessed. Fewer complications were observed in the female group that received BSM (*p* = 0.009, NNT 5.3) and in the male group treated with BSM augmentation (*p* = 0.03). The incidence of pseudarthrosis decreased in the female group with bone substitute material (*p* = 0.02, NNT 13.2). Especially among elderly female patients, there was a significant reduction in nonunions following BSM augmentation treatment (*p* = 0.001, NNT 7.5). Regardless of the treatment, necrosis and infection rates were similar among female (*p* = 0.6) and male patients (*p* = 1.0). Delayed wound healing, soft tissue tears, chronic regional pain syndrome (CRPS), inactivity-related osteoporosis, cartilage damage, previous dead, secondary dislocation, post surgery psychological disorders (e.g., delirium), neurological diseases, premature removal of metal implants, and delayed bone healing exhibited no significant differences.

Radiological evaluation of the bone healing process revealed an intriguing discovery. Both BSM and the ED group displayed accelerated bone healing processes in both female and male patients (*p* = 0.001), followed by subsequent harmonization (*p* > 0.05). Although further investigation is needed to elucidate the exact mechanism behind this finding, it suggests that bone substitute material may expedite the bone healing process in male and female patients. Notably, the age and demographic characteristics of the study population were similar between male and female patients., indicating a balanced sample distribution.

### 3.2. Comparison to Previous Studies

Consistent with prior research, this study found that female patients experienced fewer complications when BSM was used for augmentation compared with cases where the defect was left untreated. This suggests that BSM may reduce the risk of complications in female patients with traumatic fractures. Specifically, the incidence of pseudarthrosis was significantly lower in the female group that received BSM augmentation, with the difference being more pronounced in female patients over 65.

Contrary to previous studies indicating increased muscle loss in female patients [26], the findings in this study highlight the benefits of BSM augmentation for female patients following traumatic long bone fractures. Although previous preclinical studies suggested increased resorption activity in female patients and a higher rate of osteoclasts and TRAP in aged female mice [2], the clinical findings contradict this assumption. Geriatric female patients demonstrated an improved bone healing process. In contrast to this study, stress fractures in female patients have been associated with worsened fracture healing processes [15]. Furthermore, an adverse influence of hypercoagulability and hyperfibrinolysis on revascularization and the bridging process had less clinical impact in the BSM group [27]. Fracture healing in male mice appears to be faster compared with female mice [4]. Despite an already improved bone-healing process, male patients still demonstrated benefits from using bone substitute material (e.g., fracture edge *p* < 0.001). The findings in this study support an enhanced bone healing process in both male and female patients.

Fewer complications were observed in the geriatric female patient cohort with added bone substitute material with an NNT of 7.5. From the assumption of previous studies of an NNT less than 8, BSM in geriatric female patients presented benefits [28]. Regarding complications, the NNT was 5.3 patients to prevent one complication, as BSM seems to prevent a larger number of complications. In terms of infections, no differences were observed between male and female cohorts in both the empty defect and bone substitute augmentation treatments. While the male gender has been considered a risk factor for major surgical site infections, it did not significantly influence the use of augmentation [9,11]. Fewer pseudarthroses were detected in the geriatric female patient cohort with bone substitute augmentation (*p* = 0.01). Previous studies supported the assumption of risk factors for pseudarthrosis, including female sex and advanced age [29,30]. The findings of this study regarding reduced pseudarthrosis in female patients using bone substitute material align with the assumption that it supports the bone healing process following bone defect augmentation [31]. Notably, female geriatric patients exhibited an improved healing process with fewer nonunions, contrary to delayed bone healing processes observed in female geriatric mice [2].

Contrary to findings in osteoarthritis [32], female patients exhibited no differences compared with the male cohort and the bone substitute augmentation group in terms of neurological diseases such as pain or paresthesia.

In summary, this study’s results revealed an enhanced bone healing process in both male and female patients.

### 3.3. Limitation of This Study

While this study boasts notable strengths, including substantial sample size and utilization of a level one trauma center, it is imperative to acknowledge certain limitations. One limitation stems from the retrospective study design, which may introduce biases associated with the absence of randomization. Although unaware of patient inclusion in the study, surgeons may have favored using bone substitute augmentation for larger bone defects, potentially introducing bias. To mitigate this, examiners were blinded to the patient’s assignment to the BSM or ED group. Another limitation arises from the single-center structure of the study, limiting generalizability to other hospitals and compromising external validity. To enhance external validity, we strongly recommend subsequent prospective multicenter studies to validate and expand upon our findings.

Moreover, solely relying on clinical and radiological parameters as outcome measures might overlook other potential factors influencing bone healing and complications. Future studies should incorporate patient satisfaction, functional outcomes, patient-reported outcomes, range of motion, and quality of life in their assessments.

## 4. Materials and Methods

### 4.1. Group Formatting

A total of 355 patients with traumatic bone fractures of the proximal humerus, proximal tibia, or distal radius were included in this study, the observation period spanned seven years, from 2011 to 2018.

In this retrospective trial, all patients underwent surgical treatment. They were divided into two groups: one received BSM, and the other had ED without defect filling. Fractures augmented with injectable bone-substitute material were treated with either calcium phosphate (CP, Calcibon^®^ by Zimmer Biomet Deutschland GmbH, Freiburg, Germany) or nanocrystalline hydroxyapatite (NHA, Ostim^®^ by aap Biomaterials GmbH, Dieburg, Germany). Grounded in the clinical equivalence of both bone materials in previous trials, this study described the bone substitute group without differentiation. According to the criteria established in this study, data analysis included 345 patients (Figure 5).

### 4.2. Baseline Characteristics and Demographic Data

Each patient underwent a series of up to five postsurgical examinations. Exclusion criteria were applied to patients who lacked postsurgical examinations, had pathological fractures, or suffered from stress fractures (n = 10, 2.82%). Patients with substantial missing demographic data were excluded from the study (Figure 6).

To create a reference cohort with matching age and fracture severity for the control group, we ensured similarity by employing the Mann-Whitney-U test to assess the significance of differences between the groups.

### 4.3. Fracture’s Severity and Previous Illness

In this study, we conducted a comprehensive examination of fractures occurring in various anatomical regions. Specifically, we focused on fractures in the upper extremity, including proximal humerus and distal radius fractures, as well as fractures in the lower extremity, particularly proximal tibia fractures. To assess the severity of these fractures, we utilized the widely recognized fracture classification system developed by the Association for Osteosynthesis (AO). For this trial, we employed two distinct systems within the AO classification. Building upon prior research, we categorized fractures based on their proximity to the articular surface, designating them as AO Type A for extraarticular fractures, Type B for partial intraarticular fractures, and Type C for intraarticular fractures. Additionally, we meticulously recorded fracture classification data using a numeric code, enabling us to precisely determine the exact morphology and location of each fracture.

To classify comorbidities within patient groups, we employed the ASA Physical Status System and, when applicable, the Charlson Comorbidity Index [33]. We also collected demographic data for each patient, including gender, age, and body mass index (BMI).

Patients received surgical treatment utilizing the gold standard approach for each type of fracture. We incorporated two different bone substitute materials. First, Ostim^®^, an injectable nanocrystalline hydroxyapatite salt, was used in 15.9% of cases (n = 55). Second, a calcium phosphate-based bone augmentation material, Clacibon^®^, was employed to augment bone defects in 39.7% of cases (n = 137).

### 4.4. Radiological Bone Healing Process

To assess the bone healing process, patients underwent clinical and radiological examinations at up to five intervals. Building on previous trials, we evaluated the bone healing effect using established criteria for bone healing [22,23,24]. The assessment of the fracture healing process involved classifying the fracture gap, fracture tilt, and articular surface according to the German school grading system (1–5, with 1 being the best and 5 the worst). We also quantified bone substance and identified radiographic pathologies.

### 4.5. Postsurgical Complications

Complications are critical in the daily clinical practice of orthopedic and trauma surgery. Among the most significant are pseudarthrosis and nonunion, as defined by the FDA. In our study, pseudarthrosis was defined as a nonunion process persisting for more than six months (Figure 7). Additionally, we categorized inadequate bone healing occurring between six weeks and six months as malunion or delayed bone healing. Neurological complications included severe postsurgical pain persisting for more than six weeks, as well as conditions like paresthesia, hypesthesia, dysesthesias, or hyperesthesia. Long-term complications were assessed by evaluating cartilage damage, including conditions like chondrocalcinosis, post-traumatic arthrosis, or osteoarthritis. Furthermore, insufficient osteosynthesis could result in secondary dislocation. We also measured the duration of hospital stay as an indicator of the overall patient outcome.

Statistical analysis was conducted using IBM^®^ SPSS^®^. Demographic data and complications were categorized nominally, while the evaluating systems were treated as ordinal variables. Statistical significance was assessed using the Mann–Whitney U test for uneven distributions, with the level of significance set at 5%.

## 5. Conclusions

This study provides valuable insights into the gender-specific differences in treating bone defects of patients with traumatic long bone fractures using bone substitute material. The findings suggest that augmenting bone defects with substitute material may reduce the risk of complications, particularly pseudarthrosis, in female patients. Additionally, the study emphasizes the potential of BSM to accelerate the bone healing process. These findings underscore the significance of considering gender-specific factors when addressing bone defects and highlight the potential advantages of utilizing bone substitute material in female patients.

## Figures and Tables

**Figure 1 ijms-24-14232-f001:**
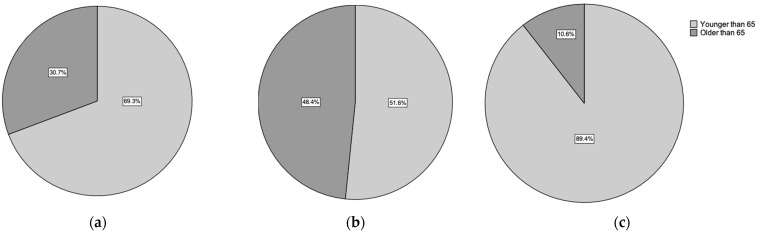
Comparing the age divided in geriatric (>65 years) and adult fractures (<65 years). Graphic of (**a**) all patients, (**b**) the female patient group, and (**c**) the male patient group.

**Figure 2 ijms-24-14232-f002:**
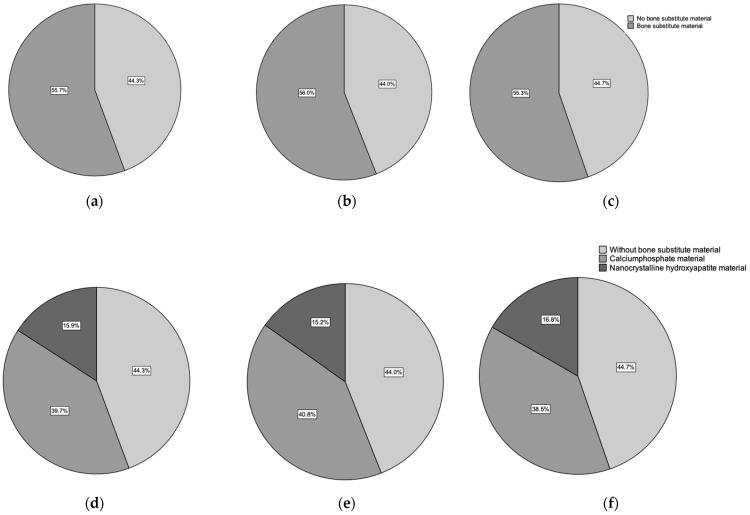
The top line shows the comparison of the groups of empty defect treatment and the treatment with bone substitute augmentation: (**a**) all patients, (**b**) the female patient group, and (**c**) the male patient group. The bottom line shows the comparison of empty defect treatment groups with the groups of calcium phosphate and the group of nanocrystalline hydroxyapatite: (**d**) all patients, (**e**) the female patient group, and (**f**) the male patient group.

**Figure 3 ijms-24-14232-f003:**
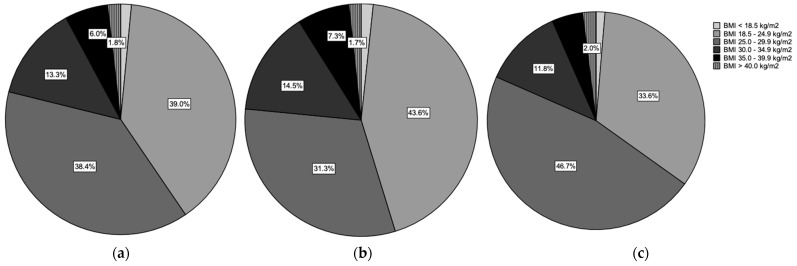
Comparing the Body Mass Index (*BMI*) using the WHO classification of obesity: (**a**) all patients, (**b**) the female patient group, and (**c**) the male patient group.

**Figure 4 ijms-24-14232-f004:**
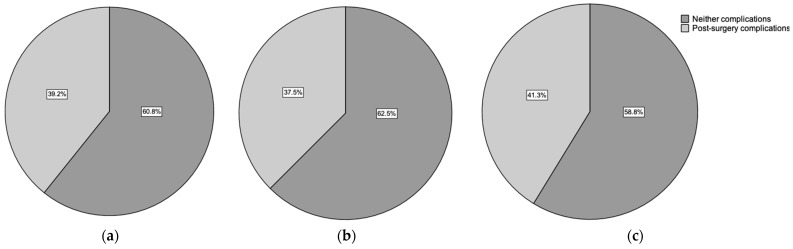
Overview of patients suffering postsurgical complications: (**a**) all patients, (**b**) the female patient group, and (**c**) the male patient group. No statistically significant differences between the groups were observed.

**Figure 5 ijms-24-14232-f005:**
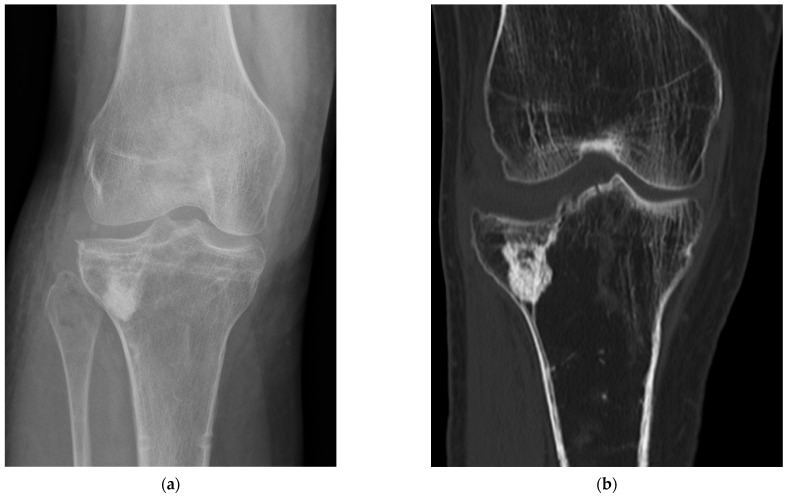
Bone defect augmentation in a proximal tibia fracture in follow-up examinations. (**a**) Assessment by conventional examination and (**b**) assessment by CT examination.

**Figure 6 ijms-24-14232-f006:**
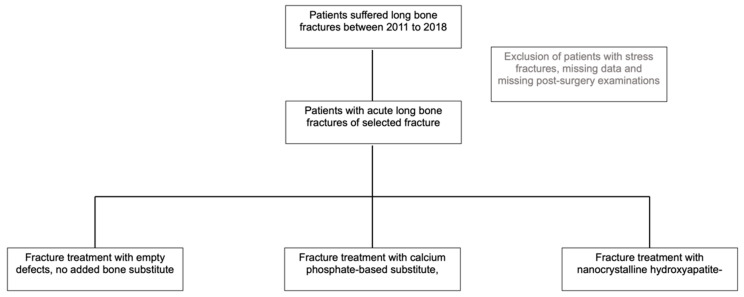
Overview of patient cohort election.

**Figure 7 ijms-24-14232-f007:**
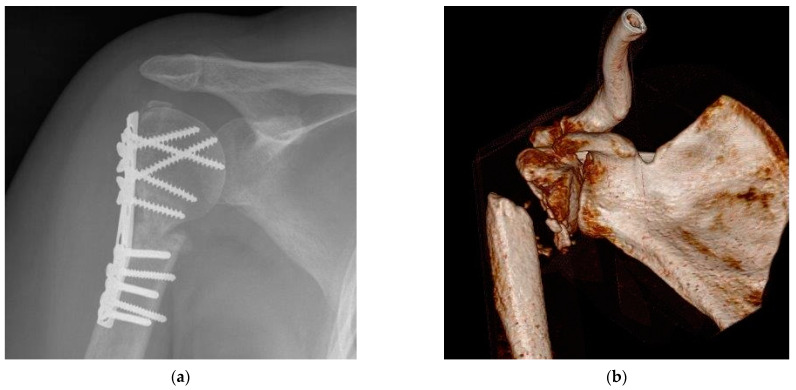
Worse postsurgical pseudarthrosis without defect void filling. (**a**) Postsurgical examination demonstrated initial treatment by osteosynthesis without bone defect augmentation. (**b**) Severe pseudarthrosis of the fracture more than a year post surgery.

**Table 1 ijms-24-14232-t001:** Previous illnesses evaluated using the ASA classification in all patients. No difference was noticed between the sex groups.

ASA Classification	Number of Patients	Relative Frequency
Missing Data	11	3.2%
1	45	13.4%
2	222	64.3%
3	66	19.1%
4	1	0.3%

**Table 2 ijms-24-14232-t002:** Overview of three analyses of complications patients suffered after surgery: (a) comparing the data between the female and the male patient group, (b) comparing the differences between the female subgroups of the BSM cohort and the ED group, and (c) comparing the differences between the male subgroup of BSM and the ED treatment group.

*p*-Value	(a) Comparing the Female and the Male Patient Cohort	(b) Female Cohort: Comparing One Defect Augmentation and Empty Defects	(c) Male Cohort: Comparing Bone Defect Augmentation and Empty Defects
Complications (yes/no)	0.4	0.009	0.026
Complications (pseudarthrosis, other complications, pseudarthrosis, and other complications)	0.5	0.018	0.027
Number of complications	0.7	0.004	0.027
Pseudarthrosis	0.6	0.022	1.0
Necrosis and infection	0.7	0.6	1.0
Belated wound healing	1.0	0.4	1.0
Soft tissue tears	1.0	1.0	0.5
Chronic regional pain syndrome	0.7	0.3	0.5
Osteoporosis by inactivity	0.5	0.5	0.3
Cartilage damage	1.0	0.049	0.5
Dead	0.5	1.0	1.0
Secondary dislocation	0.8	0.5	0.7
Psychical diseases	1.0	1.0	1.0
Neurological diseases	1.0	0.1	0.2
Premature metal removal	0.8	1.0	0.4
Belated bone healing	0.6	0.7	1.0
Duration in hospital	-	0.02	0.026

**Table 3 ijms-24-14232-t003:** The number of pseudarthroses fractures compared by treatment, gender, and age—geriatric and nongeriatric fractures (>65 years and <65 years).

	Female			Male		
Pseudarthrosis in	Empty Defect Treatment	Bone Substitute Material	*p*-Values	Empty Defect Treatment	Bone Substitute Material	*p*-Values
All ages	n = 7, 8.6%	n = 1, 1.0%	0.022	n = 4, 5.6%	n = 5, 5.6%	1.0
<65 years	n = 1, 2.4%	n = 0, 0.0%	0.4	n = 3, 4.5%	n = 3, 3.8%	1.0
>65 years	n = 6, 15.4%	n = 1, 2.0%	0.01	n = 1, 16.7%	n = 2, 18.2%	1.0

**Table 4 ijms-24-14232-t004:** Significance of postsurgical radiological bone healing criteria. Comparing the *p*-values of the bone healing process between the ED treatment group and the BSM augmentation group in the female patient cohort (left side) and male patients (right side).

Time ([Mean ± SD; Min:Max] in Days Post Surgery)	Fracture Edge		Fracture Gap		Articular Surface		Osteosynthesis	
	Female	Male	Female	Male	Female	Male	Female	Male
1.91 ± 1.21; 0:7	0.6	0.1	0.025	0.7	0.013	0.011	0.5	0.2
24.29 ± 6.39; 10:40	<0.001	<0.001	<0.001	0.002	0.002	0.002	0.6	0.9
50.91 ± 8.64; 40:70	<0.001	<0.001	<0.001	0.010	0.002	0.162	0.8	0.5
90.67 ± 15.73; 60:130	<0.001	<0.001	<0.001	0.004	0.001	0.018	0.6	0.3
247.02 ± 71.72; 135:365	0.015	<0.001	0.007	<0.001	0.006	0.1	0.6	0.3
638.15 ± 307.69; 371:1986	0.4	0.1	0.2	0.2	0.009	0.038	1.0	0.8

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
