# Peer review of "Retrospective Analysis of Bone Substitute Material for Traumatic Long Bone Fractures: Sex-Specific Outcomes"

_ijms, 2023, doi:10.3390/ijms241814232_

Round 1

Reviewer 1 Report

1. The sentences at places look incomplete , for e.g. Line 15-16 , Please check everywhere in the manuscript 

2. Need of the study is unclear

3. Introduction need to be more focussed on the need of the study 

4. Objective should be crisp and clear. Reframe 

5. The numbers in the figure are unclear 

6. Table 1 suddenly pops up from no where 

7. Result comes first and then material and methods , why ?

8. Conclusion is too long , difficult to understand what authors actually are trying to portray 

1. The sentences at places look incomplete , for e.g. Line 15-16 , Please check everywhere in the manuscript 

Author Response

The authors apologize for the elongated review time. 
We really thank you for your work. 

Reviewer 2 Report

The manuscript contained numerous English errors.

1.     “Compared with compared with”

2.      “However, female patients showed an inferior boney bridging was in.”  please check

3.       “ male rates”, should be male rats

4.       This is an observational study not a trial.

5.       2. Results should be followed by 2.1 2.2, et al

6.       “the sex 184 (53.33%) patients were female”  please revise

7.       “There was no significance between the groups of geriatric patients in the female cohort (p=1.0) and in the male cohort (p=0.45)” what do you mean no significance?

8.       Regarding Table 3 and Table 5, it's important to provide what measurements were compared. Simply reporting p-values without context can be confusing.

Too many English errors

Author Response

The authors apologize for the elongated review time. 
We really thank you for your work. 

Please look attached.

Round 2

Reviewer 2 Report

Ok, the manuscript has improved a lot.

Author Response

Dear Reviewer,

the authors thank you for your work and the proofreading. 

Sincerely, 

the authors